# Youth Exposure to Hate in the Online Space: An Exploratory Analysis

**DOI:** 10.3390/ijerph17228531

**Published:** 2020-11-17

**Authors:** Nigel Harriman, Neil Shortland, Max Su, Tyler Cote, Marcia A. Testa, Elena Savoia

**Affiliations:** 1Emergency Preparedness, Research, Evaluation, and Practice Program, Division of Policy Translation and Leadership Development, Harvard T.H. Chan School of Public Health, Boston, MA 02115, USA; masu@hsph.harvard.edu (M.S.); testa@hsph.harvard.edu (M.A.T.); esavoia@hsph.harvard.edu (E.S.); 2Center for Terrorism and Security Studies, University of Massachusetts Lowell, Lowell, MA 01854, USA; Neil_Shortland@uml.edu; 3Department of Biostatistics, Harvard T.H. Chan School of Public Health, Boston, MA 02115, USA; 4Operation 250, Lowell, MA 01854, USA; tcote@operation250.org

**Keywords:** online hate, hate speech, online disinhibition, online safety

## Abstract

Today’s youth have extensive access to the internet and frequently engage in social networking activities using various social media platforms and devices. This is a phenomenon that hate groups are exploiting when disseminating their propaganda. This study seeks to better understand youth exposure to hateful material in the online space by exploring predictors of such exposure including demographic characteristics (age, gender, and race), academic performance, online behaviors, online disinhibition, risk perception, and parents/guardians’ supervision of online activities. We implemented a cross-sectional study design, using a paper questionnaire, in two high schools in Massachusetts (USA), focusing on students 14 to 19 years old. Logistic regression models were used to study the association between independent variables (demographics, online behaviors, risk perception, parental supervision) and exposure to hate online. Results revealed an association between exposure to hate messages in the online space and time spent online, academic performance, communicating with a stranger on social media, and benign online disinhibition. In our sample, benign online disinhibition was also associated with students’ risk of encountering someone online that tried to convince them of racist views. This study represents an important contribution to understanding youth’s risk factors of exposure to hateful material online.

## 1. Introduction

Today’s youth have almost universal access to the internet and frequently engage in social networking activities using various social media platforms and devices [1]. This is a phenomenon that hate groups are exploiting when disseminating their propaganda [2,3]. Data gathered from a demographically balanced sample of over a thousand youth in the United States showed that approximately half of them, within a study period of three months, experienced exposure to hateful material while online [4]. Analyses of interviews with right-wing extremists demonstrated that communication over the internet provides an effective networking method amongst their supporters. These groups use the internet to convey their racist messages and adapt communication strategies that are appealing to youth, with the scope of recruiting new members, including the use of images, videos, music, and online games [5,6]. The online space also provides hate groups with a unique opportunity to portray an image of unity and identity [6,7]. This online collective identity self-perpetuates as a welcoming space of like-minded individuals, providing sought-after validation for potential members’ societal grievances and attracting socially isolated individuals to become members of a community that accepts them [8]. Furthermore, the anonymity of the internet creates an environment where hate groups speak or act out more radically compared to what they would do in person [9]. This is in part attributed to a psychological process referred to as online disinhibition, a process that most individuals experience when online compared to real-life leading to a lack of restraint and increased openness of expression [10,11]. The literature describes two types of online disinhibition: toxic and benign disinhibition. Toxic disinhibition manifests as a propensity towards a variety of negative attitudes and behaviors such as anger, vicious criticism, outgroup hatred, cyberbullying, racism, and aggression [12,13,14,15,16,17,18,19]. On the contrary, benign disinhibition is mostly described as a positive process by which individuals feel increased comfort in manifesting acts of kindness when online compared to in-person which, however, in some cases can lead to undesirable situations [13,20,21,22,23]. The investigation into online disinhibition and exposure to hate remains relatively nascent—to our knowledge, there has only been one study that has demonstrated an association between online disinhibition and exposure to hate online [12].

The presence, form, and function of hateful material available on the internet has been extensively discussed by academics and practitioners [24,25,26,27,28]. Past research has focused on long-term exposure to hateful content online showing that it might reinforce discriminatory views and could lead to developing defensive and hyper-vigilant attitudes [29,30,31]. Some have theorized that access to such material influences the likelihood that an individual will, eventually, engage in hateful or violent behavior [31,32,33]. However, there is limited knowledge on the risk factors that lead an individual to be exposed to online hate in the first place and on the consequences of such exposure. In recent years, the investigation into the risk factors for exposure to hate in the online space has burgeoned. In 2011, a US-based national study demonstrated that amount of general technology use and age are predictive factors for almost all technology-based violent experiences and exposures [32]. Recent research in the US has identified various socio-demographic, psychological, and behavioral factors that may put an individual at risk of exposure to hate online, such as young age, white race, male gender, level of education, online victimization, low trust in government, and time spent on the internet [34,35]. Such risk factors provide invaluable information on a pernicious phenomenon that has persisted at high levels over the past decade in the US. Indeed, data from the US have consistently indicated high levels of exposure to hate messages online among nationally sampled populations, with 53%, 65%, and 87% of respondents indicating exposure to such messages in 2013, 2015, and 2016, respectively [34,35,36]. The international knowledge base of the correlates for exposure to hate online has similarly grown over the past decade. Research from Finland has demonstrated that exposure to hate is associated with young age, social trust, weak family attachment, time spent online, social media use, visiting risky websites, offline victimization, and the production of hate messages [29,37]. Data in this context have also indicated high levels of online exposure to hate messages, with 38% and 48% of sampled respondents indicating exposure in 2013 and 2016, respectively [29,37]. Additional research from four countries (UK, Finland, US, Germany), has also supported the notion that high levels of exposure to online hate persist cross-nationally [36]. As informative as this research base is, to our knowledge, no studies in the US have attempted to identify predictors of exposure to hate messages using data from after 2016. With such high levels of exposure both in the US and internationally, it is necessary to keep the information on risk factors up to date.

Despite the growing body of literature surrounding the prevalence and risk factors for exposure to online hate internationally, to date, most studies have examined witnessing hate as either a victim—whereby the messages are targeted towards one’s own group; or as a bystander—whereby, the messages are targeted towards a different group [29,38]. Interestingly, there exists a dearth of literature that focuses on the interactions between those espousing hateful messages and those who are exposed to them. While there is extant research on the efficacy of persuasive racist narratives on youth, to our knowledge no studies have identified the risk factors associated with encountering those who attempt to persuade youth by utilizing such narratives—referred to in the literature as persuasive storytellers [31,39]. Our study aims to address this gap in the literature by identifying factors that exist upstream of the interactions between youth and persuasive storytellers, which can inform educational initiatives that focus on preventing the interaction with them in the first place, rather than discrediting their narratives after youth have been exposed to them.

There is concern that the internet may provide youth with a gateway to online hate communities and expose them to a dizzying array of sites containing hateful material and individuals espousing hatred [40]. As such, it is important to understand who are most likely to experience such exposure, to equip them with the requisite knowledge to critically assess the material they may come across while online. Careful attention must also be given to identifying various factors that may place an individual at greater risk for encountering someone who attempts to persuade them of hateful views, to provide educational initiatives opportunities to prevent this encounter from occurring. To this end, the present study aims to contribute to the growing body of literature on online hatred by attempting to elucidate the various risk factors that expose youth to hateful messages and those who espouse them in the online space. The research objectives for this exploratory study were as follows:
Identify youth demographic characteristics, behaviors, and psychological processes associated with exposure to hate messages online.Identify youth demographic characteristics, behaviors, and psychological processes associated with encountering someone trying to convince the respondent of racist views.

## 2. Materials and Methods

We implemented a cross-sectional study design, using a quantitative questionnaire, in two high schools in Massachusetts (USA). The questionnaire was implemented in December 2018 in one high school and in April 2019 in the other. At both schools, the questionnaire was implemented on paper during school hours as part of Health Class. Prior to implementation, the questionnaire was pilot tested in Utah and Massachusetts, and questions were revised based on youth feedback. A copy of the questionnaire can be found as a Appendix A. For the analyses presented in this study, we explored youth demographic characteristics, online behaviors, psychological processes, and their association with online exposure to hateful content—no other outcome variables were considered for this study. Below we present the independent and dependent variables considered in this study, as well as the methods for data analysis. Parents were provided with information on the study and opt-out forms one month prior to data collection. Consent to participate in the study was obtained from the students prior to responding to the questionnaire. The Helsinki ethics protocol was followed throughout the course of this study [41]. The study protocol and the survey questions were deemed exempt by the Harvard T.H. Chan Institutional Review Board (IRB16-1757) as well as by the ethical committees of the school districts where the study was implemented. Below, we describe the socio-demographic characteristics of the participants. Following, we provide descriptive statistics of the sample and results from simple and multiple regression models.

### 2.1. Participants

In this study, we gathered data from a convenience sample of 320 students, aged between 14 and 19 years old (students from grades nine to twelve, respectively), the majority of whom were female (58%). The sample primarily included students from grades 9 (27%) and 10 (66%), 4% were in 11th grade, and 3% in 12th grade. Major categories of race were distributed as follows: white (35%), mixed-race (21%), Hispanic (18%), and Black (8%). Regarding academic performance, the majority of students reported receiving grades ranging between A-/B+ (36%), and the fewest amount of students reporting receiving grades of C or lower (11%).

### 2.2. Independent Variables

Respondents were asked about the amount of time they spend using technology each day, which social media tool (i.e., YouTube, Snapchat, Instagram, etc.) they use, and how frequently they use it, with answer options ranging from 1 (never) to 6 (all the time). As social media use across platforms is correlated, an overall social media use variable was created by a summative score created by converting the distribution of frequency of use for each social media platform into a normal standardized distribution and adding the resultant scores across all social media platforms [42,43]. To assess which social media platforms were related to the outcomes, a Spearman’s correlation matrix was created. Respondents were asked how many of their social media followers they knew in person, and if they had recently removed any strangers from such followers; similar questions were asked about respondents’ friends’ social media behaviors, under the assumption that they would be less likely to misreport friends’ habits compared to their own. Respondents were asked to report how frequently they played online video games, and if they chatted with strangers while doing so.

Online disinhibition was measured using the online disinhibition scale after adapting the questions to the young age of the study population [13]. The factor structure of the questions was assessed with a factor analysis using principal component analysis for factor extraction, and as a result, a scale with a score ranging from 7 to 28 was formed, with higher values indicating a more disinhibited behavior. Participants’ perception of online risk was measured by asking the respondent to rate the risk of seven online scenarios. In this case as well, a factor analysis was performed to assess the structure of the questions, and as a result, a scale was created with scores ranging from 7 to 35, with lower values indicating lower risk perception. For both scales, Kaiser–Meyer–Olkin (KMO) measure of sampling adequacy and Bartlett’s test of sphericity were used to test for the suitability of the data for factor analysis. Cronbach alpha was computed to assess the scales’ reliability. Questions about engagement in risky online behaviors were also asked, such as chatting with strangers or sharing personal information with online contacts. Finally, respondents were asked about parents’ supervision of their online activities and if they had a trusted adult to ask for help if they encountered an uncomfortable situation online.

### 2.3. Dependent Variables

Two dependent variables were created to measure exposure to online hate during the two months prior to the survey. Respondents were asked to report how frequently they had come across insulting verbal or written expressions against a specific group because of their race, religion, disability, sexual orientation, ethnicity, gender, or gender identity (*exposure to online hate messages*). Respondents were also asked if, in the same period, they encountered *someone trying to convince them of racist views*. Both variables were dichotomized as yes/no.

### 2.4. Statistical Analyses

Simple and multiple logistic regression models were used to study the association between the independent and dependent variables. A Box-Tidwell procedure was used to confirm that the benign disinhibition score, the only continuous variable in the model, had a linear relationship to the log odds of each dependent variable. Independent variables were included in the multiple models when a statistically significant association was found in the simple models (*p*-value < 0.05). Gender, race, and school year were included regardless of their significance in the final models because of theoretical relevance. Hosmer–Lemeshow tests were used to assess the goodness of fit of the models. Race and gender were also tested as interaction terms [44]. Data analysis was performed using Stata Statistical Software: Release 15.1. College Station, TX: StataCorp LLC.

## 3. Results

We provide below the sample characteristics and descriptive statistics for social media use and online behaviors, adult supervision of online activities, and exposure to hate online. Following, we present the results of the factor analyses for the online disinhibition and risk perception scales, and the results of the simple and multiple logistic regression models for exposure to online hate.

### 3.1. Sample Characteristics and Descriptive Statistics

The majority of the respondents (75%) reported spending over three hours a day interacting with technology, excluding schoolwork. YouTube, Snapchat, and Instagram were the most frequently used social media tools with over 90% of respondents reporting their use. Only 7% of the respondents knew all of their social media followers, the majority of them (63%) reported that in the two months prior to the survey, they chatted with someone on social media that they had not met in person and 45% reported they believe their friends did so as well. During the same timeframe, 67% removed a follower from their social media account they had not met in person, and 48% reported to have shared personal information, such as their school or town name when posting on social media. Among those playing video games online, 52% reported that in the two months prior to the survey they chatted with someone they did not know while gaming. Regarding parents’ supervision, the majority (57%) of students had parents that occasionally asked about their online activities, but only 25% reported that their parents had rules for what they did online and checked on them to make sure they followed the rules. The majority (64%) of students reported they had a trusted adult they could ask for help if they experienced an online situation that made them feel uncomfortable. More details on the students’ online activities are provided in Table 1.

Fifty-seven percent of students reported to have come across hate messages on social media or on a website in the two months prior to the survey and twelve percent reported to have encountered someone online that tried to convince them of racist views during the same time period. Details on students’ exposure to hate online is provided in Table 2.

In Table 3, we report the correlation between social media platforms and hate online. Exposure to hate messages online was significantly correlated with Twitter use (*r* = 0.12, *p* < 0.05) and Houseparty use (*r* = 0.12, *p* < 0.05). Encountering someone trying to convince the respondent of racist views was only significantly associated with YouTube use (*r* = 0.12, *p* < 0.05).

### 3.2. Factor Analyses and Descriptive Statistics of Risk Perception and Online Disinhibition

#### 3.2.1. Risk Perception Scale

Bartlett’s Test of Sphericity (*χ*^2^ = 865.397, *df* = 21, *p <* 0.01) and Kaiser–Meyer–Olkin measure of sampling adequacy (0.834) indicated that the data on the seven questions designed to measure risk perception were suitable for factor analysis. A factor analysis was computed resulting in one factor with an eigenvalue greater than one, all items had a factor loading greater than 0.4, and 52% of the variance in the data was explained by the model. Cronbach’s alpha was 0.84. Risk perception scores were negatively skewed with a mean of 29.3 (SD = 5) and a median of 31 (range: 7 to 35). In the simple and multiple models, risk perception was examined as a dichotomous variable. Individuals with scores less than or equal to 28 (25th percentile) were defined as having “low risk perception” (n = 103). Individuals with scores greater than 28 were defined as having “high risk perception” (n = 217).

#### 3.2.2. Benign Disinhibition Scale

Bartlett’s Test of Sphericity (*χ*^2^ = 553.869, *df* = 55, *p <* 0.01) and Kaiser–Meyer–Olkin measure of sampling adequacy (0.8) indicated that the data on the online disinhibition questions were suitable for factor analysis. A factor analysis was computed on the 11 questions resulting in two factors with eigenvalues greater than one which explained 45% of the variance in the data. After oblique promax rotation, one question with a factor loading below 0.4 (q7f—see Appendix A) was discarded. The resulting two subscales were similarly structured to the scale developed by Udris et. al. [13] The benign disinhibition subscale contained seven questions (7a, b, d, e, g, h, k—see Appendix A) and had a Cronbach’s alpha of 0.72. The toxic disinhibition subscale contained three questions. Due to its low internal consistency (Cronbach’s alpha = 0.53), toxic disinhibition was not included in the simple and multiple regression analyses. Benign online disinhibition scores had a mean of 18.4 (SD = 4.7) and a median of 18 (range: 7 to 28). Box–Tidwell test results from the simple regression analyses supported the assumption that the distribution of benign online disinhibition scores was linear to the log odds of respondents’ exposure to hate messages online (*p =* 0.4) and to the experience of encountering someone online that tried to convince them of racist views (*p* = 0.8). Descriptive statistics of the risk perception and benign disinhibition scale can be found in Table 4.

### 3.3. Simple Models

#### 3.3.1. Exposure to Hate Messages Online

In the simple regression models, the following variables were significantly associated with the dependent variable - exposure to hate messages online: time spent online (OR = 2.3, 95% CI 1.4–3.8), removing a follower from a social media account - whom the respondent had not met in person (OR = 1.9, 95% CI 1.2–3), communicating with someone on social media that the respondent had not met in person (OR = 2.1, 95% CI 1.3–3.3), presence of parental rules for online activities (OR = 1.6, 95% CI 1–2.5), benign online disinhibition (OR = 1.09, 95% CI 1.04–1.14), and good academic performance (OR = 2.3, 95% CI 1.1–4.6). Detailed results for the simple models of exposure to hate messages online can be found in Table 5.

#### 3.3.2. Encountering Someone Trying to Convince the Respondent of Racist Views

In the simple regression models, the following variables were significantly associated with the dependent variable—encountering someone online that tried to convince the respondent of racist views: school year (OR = 0.5, 95% CI 0.2–0.9), playing video games online (OR = 2.1, 95% CI 1–4.4), and benign online disinhibition (OR = 1.19, 95% CI 1.09–1.29). The only significant categorical predictor with more than two categories was chatting—while gaming online—with someone the respondent had never met in person (*p =* 0.041). Detailed results for the simple models of exposure to hate messages online can be found in Table 5.

### 3.4. Multiple Models

#### 3.4.1. Exposure to Hate Messages Online

The overall Likelihood Ratio (LR) chi–square test statistic for the multiple logistic regression model exploring the association between the independent variables and exposure to hate messages online was significant (*χ*^2^ = 40.54, *df* = 9, *p <* 0.01). Hosmer–Lemeshow Goodness of Fit test results confirmed that the model was a good fit for the data (*χ*^2^ = 6.62, *df* = 8, *p =* 0.578). Time spent online was associated with increased odds of exposure to online hate messages—youth that spent three or more hours a day online had 2.4 times the odds of reporting exposure to hate messages (seen either on a website or social media), (OR = 2.4, 95% CI 1.3–4.3) compared to those who spent less than 3 h a day online. The odds of reporting exposure to hate messages among those who communicated with someone on social media that they had not met in person were 1.7 times that of those who had not done so (OR = 1.7, 95% CI 1–2.9). Benign online disinhibition was associated with reporting exposure to hate messages online—each one-unit increase in score on the benign disinhibition scale resulted in a 6% increase in the odds (OR = 1.06, 95% CI 1–1.12) of reporting exposure to such messages. Good academic performance was also associated with exposure to online hate messages, students who reported receiving grades greater than a C had 3.4 times the odds of reporting exposure to such messages compared to students who were receiving Cs or lower grades (OR = 3.4, 95% CI 1.5–7.7). Gender and race were used to study their interaction with the following variables: time spent online, benign disinhibition, and communicating with a person not met in person while online. None of these interactions resulted to be significant. Detailed results for the multiple models of exposure to online hate messages can be found in Table 6.

#### 3.4.2. Encountering Someone Trying to Convince the Respondent of Racist Views

The overall LR chi–square test statistic for the model investigating the association between independent variables and encountering someone trying to convince the respondent of racist views was significant (*χ*^2^ = 26.36, *df* = 7, *p <* 0.01). Hosmer–Lemeshow Goodness of Fit test results confirmed that the model was an appropriate fit for the data (*χ*^2^ = 6.13, *df* = 8, *p =* 0.6325). Only benign online disinhibition was associated with students’ risk of encountering someone trying to convince the respondent of racist views. Each one-unit increase in benign disinhibition score resulted in a 19% increase in the odds (OR = 1.19, 95% CI 1.09–1.31) of experiencing this situation. There were no significant interaction terms observed. Detailed results can be found in Table 6.

## 4. Discussion and Conclusions

Technology has become increasingly important in the lives of adolescents who are heavy users of various forms of electronic communication such as instant messaging, e-mail, social media, and sites where they share opinions, photos, and videos [45]. Although teens usually find valuable educational support and information on the internet, they can also be exposed to negative influences, such as online propaganda, hate messages, and racism. The work presented in this manuscript aimed to explore some of the risk factors for such negative online influences. Specifically, our goals were to identify youth demographic characteristics, behaviors, and psychological processes associated with online exposure to hate messages and encountering someone trying to convince the respondent of racist views.

In our study, we found that the more time youth spent online the more likely they were to be exposed to hate in the online space. This result is consistent with the previous literature [29,34,35]. Not surprisingly, communicating with strangers online was associated with an increased risk of being exposed to hate. Interestingly, good academic performance was also associated with increased risk of exposure, this may be due to increased awareness and ability to recognize the online material as hateful or by some interest in the topic expressed by higher educated youth, as found in previous research [35]. Finally, our data indicate that the more individuals felt disinhibited online, “loosened up, felt less restrained, and expressed themselves more openly”, the more likely they were to be exposed to hateful propaganda and to encounter individuals attempting to convince them of racist views. Our results raise interesting questions about the nature of disinhibition and the underlying processes that moderate the relationship between disinhibition and exposure to hate. What is of specific interest here, however, is that benign disinhibition was significantly associated with exposure to hateful material. We believe that while toxic disinhibition may be associated with active engagement in hate and harmful activities, benign disinhibition as a whole may be associated with passive exposure to hate [12]. These preliminary results pose intriguing and critical questions for future research. First, does someone exhibiting only benign disinhibited behaviors stray free from the innate risks of hateful content? Second, what role does mere exposure play on one’s own future behavior, feelings, and psychology?

The results from this exploratory study can play a useful role in the development of educational interventions and programs that aim to prevent youth from encountering individuals espousing hateful views and equip youth with the knowledge they need to appropriately react to such material. In many prevention spheres, prevention efforts have focused on educating about what risks are present online, and indeed the prevalence and nature of nefarious online actors and groups. Our results pose important implications for efforts to increase online safety, in that educational initiatives need to prioritize self-reflection and self-awareness as much as content-based knowledge. Preliminary studies into online safety education programming show that initiatives focused only on enhancing knowledge might be missing the spot for long-term education [46]. In an evaluation of existing internet safety resources and programs, results have shown that experiencing a risky online situation is not about “lack of knowledge,” but rather omission of the necessary skills, referred to as digital literacy, needed when navigating the internet [47]. Furthermore, existing literature in the European context, has demonstrated that there is a gap in the digital literacy of adolescents, especially concerning the ability to critically assess the credibility of information they encounter online [48]. This is particularly relevant to the current study, as the use of misinformation and the propagation of racialized myths—referred to in the literature as persuasive storytelling—is central to the internal communication and recruitment strategies of hate groups [24,31,39,49]. Past research has recommended that school-based education programs aimed at improving digital literacy and online safety should include the promotion of “soft competencies,” including the development of knowledge, attitudes, and skills to critically and safely experience the online space [50]. Based on the results of the current study, such competencies may include learning how to limit one’s own time online, self-awareness of disinhibited behavior, and avoiding risky situations, such as engaging with strangers.

Interestingly, these requisite competencies for safe online activity may not be limited to adolescents. Previous research on a sample of parents of high school students indicated that on average, parents were only able to correctly answer half of the questions related to the ability to assess the credibility of online situations and have safe interactions with strangers [51]. Other studies focused on high school teachers show similar weaknesses in their ability to assess the credibility of information found online, which is part of an overall lack of digital literacy [52,53]. Informed adults are a critical component of improving the digital literacy of students [52]. As such, when attempting to improve the digital literacy and online safety of students, it is necessary to focus on not only the students’ competencies but also that of the influential adults in their lives—their parents and teachers. Future research should explore the relationship between the digital literacy competency of students and influential adults in their lives, including their parents and teachers.

When reflecting on the findings of this study, it is imperative to recognize its limitations. The primary limitation is its cross-sectional design. As such, the observed relationship between independent and dependent variables, albeit plausible, should not be assumed as causal. Additionally, due to the cross-sectional nature of these data, the direction of the observed relationship between exposure and outcome can only be speculated. As some of the exposures and outcomes could be perceived as negative, there is potential for social desirability bias to influence students’ responses. As a non-random convenience sample, it is important to acknowledge that these results are not necessarily generalizable outside of the sample, and that selection bias may have also influenced the results. The reported study results are also limited in their generalizability due to the relatively small size of the sample from which they were obtained. Regarding social media use, we believe there are limitations to measuring the effect of individual platforms. As the use of individual social media platforms was correlated, we did not feel it was appropriate to include them for investigation in the regression models we generated. While our results indicated that Twitter and Houseparty were associated with exposure to hate online (*r* = 0.12), and YouTube was associated with exposure to an individual trying to convince the respondent of racist views (*r* = 0.12), we decided against including social media platforms as individual predictors in our models due to concerns of collinearity (Table 3). Future research should more systematically investigate the influence of individual platforms on exposure to hate online. Finally, there may be limitations with the measurement of online disinhibition within our sample due to the lack of variance explained by the scale and the fact that toxic disinhibition could not be reliably measured.

Similarly to previous research, our study of a population of high school students found an association between exposure to hate messages in the online space and time spent online, academic performance, communicating with a stranger on social media, and benign online disinhibition. In our sample, benign online disinhibition was also associated with students’ risk of encountering someone online that tried to convince them of racist views. This study represents an important contribution to understanding youth’s risk factors of exposure to hateful material online. These findings can lead to important preliminary recommendations for the development of educational activities aimed at improving online safety that should focus on teaching youth how to limit the time they spend online, helping them recognize the disinhibition effect of the internet and how passive online behaviors may also generate risk and influence decisions they make when online. Based on the existing literature, we recommend that future initiatives that focus on preventing youth exposure to online hate also incorporate a component for parents and teachers, as these adults play a critical role in how youths behave online. Finally, the results from this exploratory study can contribute to an actionable knowledge base of the determinants of exposure to hateful online material by providing the basis for future longitudinal investigation and hypothesis testing of the impact of online behaviors and novel psychological processes, including online disinhibition, on the frequency of exposure to hate messages and those who espouse them.

## Figures and Tables

**Table 1 ijerph-17-08531-t001:** Students’ Online Activities.

**Friends Communicated with Strangers (Q12)**	**N (%)**	**Shared Personal Information Online (Q11)**	**N (%)**
I am not sure	78 (24%)	I do not have a social media account	7 (2%)
Never	24 (8%)	Never	161 (50%)
Sometimes	143 (45%)	Sometimes	114 (36%)
Often	75 (23%)	Often	38 (12%)
**Social media followers met in person (Q8)**	**N (%)**	**Chatted with strangers while gaming (Q14)**	**N (%)**
I am not sure how many	35 (11%)	I do not play video games	71 (22%)
Some of them	67 (21%)	Never	120 (38%)
Most of them	188 (60%)	Sometimes	76 (24%)
All of them	22 (7%)	Often	53 (17%)
**Time spent online (Q6)**	**N (%)**	**Trusted adult (Q18)**	**N (%)**
Less than 1 h	7 (2%)	No	34 (11%)
More than 1 but less than 3 h	74 (23%)	Yes	203 (63%)
3 to 6 h	155 (49%)	Not sure	56 (18%)
More than 6 h	83 (26%)	It depends on the situation	27 (8%)
**Communicated with strangers on social media (Q10)**	**N (%)**	**Parental rules for online activity (Q16)**	**N (%)**
I do not have a social media account	6 (2%)	Do not have rules	130 (41%)
Never	111 (35%)	Have a few rules but do not check to see if they are followed	111 (35%)
Sometimes	156 (49%)	Have a few rules and check	67 (21%)
Often	46 (14%)	Have many rules and check	11 (3%)
**Parents ask about online activities (Q15)**	**N (%)**	**Play video games online (Q13)**	**N (%)**
Parents do not ask	71 (22%)	Never	141 (44%)
Parents occasionally ask	181 (57%)	Sometimes	95 (30%)
Parents frequently ask	49 (15%)	Often	84 (26%)
I think my parents check on my devices	12 (4%)	**Removed strangers from social media followers (Q9)**	**N (%)**
I am never online	6 (2%)	No	105 (33%)
		Yes	215 (67%)

**Table 2 ijerph-17-08531-t002:** Students’ Exposure to Hate Online.

Exposed to Hate Messages Online (Q22—Social Media or Website)	N (%)	Encountered Someone Trying to Convince the Respondent of Racist Views (Q17)	N (%)
Yes	182 (57%)	Yes	38 (12%)
No	138 (43%)	No	282 (88%)

**Table 3 ijerph-17-08531-t003:** Spearman’s Correlation Matrix for Social Media and Hate Online

	1	2	3	4	5	6	7	8	9	10	11	12	13	14	15	16
**1. Hate messages**	−	−	−	−	−	−	−	−	−	−	−	−	−	−	−	−
**2. Racist Views**	0.16 **	−	−	−	−	−	−	−	−	−	−	−	−	−	−	−
**3. Skype**	0.04	0.09	−	−	−	−	−	−	−	−	−	−	−	−	−	−
**4. Flickr**	0.00	−0.03	−0.01	−	−	−	−	−	−	−	−	−	−	−	−	−
**5. Kik**	−0.02	−0.09	0.04	0.01	−	−	−	−	−	−	−	−	−	−	−	−
**6. Twitter**	0.12 *	0.03	0.07	0.11	−0.06	−	−	−	−	−	−	−	−	−	−	−
**7. Facebook**	0.03	−0.05	−0.05	0.04	0.11	0.12 *	−	−	−	−	−	−	−	−	−	−
**8. Houseparty**	0.12 *	−0.01	−0.07	−0.01	0.05	0.03	−0.04	−	−	−	−	−	−	−	−	−
**9. YouTube**	−0.02	0.12 *	0.16 **	−0.04	−0.04	0.09	−0.06	0.00	−	−	−	−	−	−	−	−
**10. WhatsApp**	−0.08	−0.03	0.02	0.05	0.10	0.04	0.14 *	−0.02	0.03	−	−	−	−	−	−	−
**11. Instagram**	0.04	0.03	0.06	−0.07	−0.02	0.02	0.04	0.07	0.18 **	0.03	−	−	−	−	−	−
**12. Pinterest**	0.00	−0.03	0.01	0.04	0.06	0.05	0.16 **	−0.05	−0.07	0.17 **	0.15 *	−	−	−	−	−
**13. Snapchat**	0.07	0.04	−0.05	−0.03	−0.01	0.05	0.13 *	0.18 **	0.11	−0.13	0.55 **	0.13 *	−	−	−	−
**14. TikTok**	0.04	−0.05	0.03	0.01	−0.01	0.13 *	0.03	0.04	−0.04	−0.01	0.10	0.08	0.06	−	−	−
**15. Google+**	0.00	−0.02	0.04	0.08	0.05	−0.12 *	0.04	0.07	0.05	0.05	0.13 *	0.15 **	0.10	−0.05	−	−
**16. VSCO**	−0.05	0.04	−0.05	0.13 *	0.01	0.11 *	0.17 **	0.00	−0.11	0.05	0.33 **	0.28 **	0.32 **	0.04	0.03	−

* *p* < 0.05; ** *p* < 0.01.

**Table 4 ijerph-17-08531-t004:** Risk Perception and Benign Disinhibition Scales

Risk Perception Scale (Q19)		Benign Online Disinhibition Scale (Q7a, b, d, e, g, h, k)	
Mean	29.35	Mean	18.44
Standard Deviation	4.99	Standard Deviation	4.72
Median	31	Median	18
Range	7–35	Range	7–28

**Table 5 ijerph-17-08531-t005:** Simple Models

Independent Variables	Exposed to Hate Messages Online	Encountered Someone Trying to Convince the Respondent of Racist Views
**Dichotomous and Continuous Predictors**	**OR (95% CI)**	
School year (10th grade and above vs. 9th grade)	0.8 (0.5–1.3)	0.5 (0.2–0.9) *
Race (White vs. Non-white)	1 (0.6–1.7)	1.5 (0.8–3.1)
Gender (Female vs. Male)	1.5 (0.9–2.3)	0.7 (0.3–1.3)
Time spent online (≥3 h vs. <3 h)	2.3 (1.4–3.8) **	1.6 (0.7–3.8)
Removed a stranger from your social media followers (Yes vs. No)	1.9 (1.2–3) *	2 (0.9–4.5)
Communicated with strangers on social media (Sometimes/Often vs. Never/No social media account)	2.1 (1.3–3.3) **	2 (0.9–4.4)
Shared personal information online (Sometimes/Often vs. Never/No social media account)	1.3 (0.8–2)	1.4 (0.7–2.8)
Played online video games (Sometimes/Often vs. Never)	1.2 (0.8–1.8)	2.1 (1–4.4) *
Parents have rules for what you do online (Any level of rule vs. Do not have rules)	1.6 (1–2.5) *	0.5 (0.3–1)
Parents ask what you do online (Any frequency of asking vs. Do not ask)	1.5 (0.9–2.6)	1.7 (0.7–4.1)
Benign Online Disinhibition	1.09 (1.04–1.14) **	1.19 (1.09–1.29) **
Risk perception (≤25th Percentile vs. >25th Percentile)	1 (0.6–1.7)	1.1 (0.5–2.3)
Academic performance (≥C+ vs. ≤C)	2.3 (1.1–4.6) *	1.1 (0.4–3.3)
Followers met in person (Not sure/Some of them vs. Most of them/All of them)	1.3 (0.8–2)	1.2 (0.6–2.6)
**Categorical Predictors with 3+ Categories**	**Chi-Squared (*df*); *p-*value**	
Friends communicate with someone on social media they had not met in person	6.92 (3); *p-*value = 0.075	5.69 (3); *p-*value = 0.128
Chatting with someone you have never met while gaming online	0.14 (2); *p-*value = 0.933	6.39 (2); *p-*value = 0.041
Having a trusted adult to speak to in case you come across something unsafe online	0.37 (3); *p-*value = 0.946	4.89 (3); *p-*value = 0.180
Social media use quartiles	1.07 (3); *p-*value = 0.785	1.06 (3); *p-*value = 0.786

* *p* < 0.05; ** *p* < 0.01.

**Table 6 ijerph-17-08531-t006:** Multiple Models

Independent Variables	Exposed to Hate Messages Online	Encountered Someone Trying to Convince the Respondent of Racist Views
**Dichotomous and Continuous Predictors**	**OR (95% CI)**	
School year (10th grade and above vs. 9th grade)	0.8 (0.5–1.5)	0.6 (0.3–1.4)
Race (White vs. Non-white)	1.1 (0.6–1.8)	1.9 (0.9–4)
Gender (Female vs. Male)	1.5 (0.9–2.5)	1 (0.4–2.7)
Time spent online (≥3 h vs. <3 h)	2.4 (1.3–4.3) **	NA
Removed a stranger from your social media followers (Yes vs. No)	1.6 (0.9–2.7)	NA
Communicated with strangers on social media (Sometimes/Often vs. Never/No social media account)	1.7 (1–2.9) *	NA
Played online video games (Sometimes/Often vs. Never)	NA	1.2 (0.4–3.8)
Parents have rules for what you do online (Any level of rule vs. Do not have rules)	1.4 (0.8–2.3)	NA
Benign Online Disinhibition	1.06 (1–1.12) *	1.19 (1.09–1.31) **
Academic performance (≥C+ vs. ≤C)	3.4 (1.5–7.7) **	NA
**Categorical Predictors with 3+ Categories**	**Chi–Squared (*df*); *p-*value**	
Chatting with someone you have never met while gaming online	NA	0.52 (2); *p-*value = 0.77

* *p* < 0.05; ** *p* < 0.01.

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
