# Peer review of "Youth Exposure to Hate in the Online Space: An Exploratory Analysis"

_ijerph, 2020, doi:10.3390/ijerph17228531_

Round 1

Reviewer 1 Report

Dear authors,

I read your article very carefully and with interest. Unfortunately the manuscript presents format and structure issues that can be significantly improved. Find below major points in the article which needs clarification, refinement, reanalysis, rewrites and/or additional information and suggestions for what could be done to improve it.

Introduction (Section 1)

From the section 1 (introduction) the aim and / or objectives of the study and / or hypotheses and / or research questions are absent or unclear, and which should be numbered.

To help you, here are some questions:

What is the importance of making this study / contribution that it brings to the literature in the field?

Why should readers be interested?

What problem/ gap resolve/fill this research?

To fill this gap (resolve this problem) what solution / intervention / benefits does this research bring? (in other words, how the proposed study will remedy this deficiency/gap/problem and provide a unique contribution to the literature).

What is the research question which address to the purpose of the research?

Materials and Methods (Section 2)

Section 2 of the introduction (first paragraph / lines 63-75), the type of methodology you used is missing. I have understood the methodology while reading the manuscript, but someone else who will read it may not understand it. So you need to be clear.

You also write that the questionnaire is in the appendix (line 69), but there is no appendix in your manuscript.

If I consider the questionnaire to be the questions presented in table 1, then your manuscript should be completely revised. Most of the questions that are asked focus on perceptions and online behaviors.  Reading your manuscript I understand that you want to focus mainly on the questions mentioned (a) “Exposed to hate messages online” and (b) “Encountered someone trying to convince the respondent of racist views”. If this is the case, then the research you present in your manuscript will be secondary, which uses some of the primary data for analysis and presentation, from the research written in the section 2 (lines 63-69). If you decide to use all the raw data in your manuscript, then you have to make significant changes and add missing things, eg for the pilot survey, where you found the questionnaire, etc.

Lines 81-84 indicate that total use of social media has been grouped. It would be really interesting if you could give more information, eg through the correlation process to present which platform seems to be most at risk.

Logistic regression models you used is ok with me.

Results (Section 3)

Usually the demographics presented in percentages are not presented in a table, only a reference is made in the manuscript.

If you decide to keep table 1 when reviewing your manuscript, remove questions that show only percentages (because these will be written in the text). Also, the questions based on table 1 (as well as all relevant questions if any - did not attach the questionnaire to get a complete picture), e.g. Q6, Q8, Q10, Q11, Q12, Q13, Q14 and Q18, it would be good to present the minimum and maximum values (if they exist), mean and standard deviation.

Discussion (Section 4)

The discussion section needs to be revised, as it looks a bit messy now. Please highlight the main points, connecting them to your results, and state why those are important and contribute to the current debate on the topic. I recommend to clarify this in text.

Also, I suggest to clearly evidence the implications for research, practice and/or society.

Conclusions (Section 5)

Υou will have to write new conclusions, because now they look a little poor.

Other remarks / comments / suggestions:

  • The first paragraph in section 4 (discussion) is a literature review. You could move this paragraph, but in a smaller format, in section 1 (introduction). If these are considered to be fully referenced in your manuscript in order to be supported after your results, then it would be good after section 1 to create a literature review section. If this is done, then the introduction will include only the aim, objectives, etc.
  • For your convenience, you can merge the discussion and conclusions sections into one section (discussion and conclusions).
  • If this research is a follow-up to a previous research or is part of a larger research, this should be clearly stated in the introduction and not later.
  • Based on the data presented in table 1 you could make a further analysis and other interesting results can be obtained, for example if there is a difference between the two schools, the genders etc or in which social media they use more etc.
  • After the revision of your manuscript you may need to change the title.

Author Response

Hello,

We would like to begin by sincerely thanking you for taking the time to provide us with such constructive and detailed feedback in your comments. We believe that by implementing your very helpful feedback, we have produced a clearer and more informative manuscript. Please see the attached document for a point-by-point response to your comments.

Reviewer 2 Report

The article presents the results of an interesting study that analyzed exposure to hate messages and the search for someone who tries to convince the respondent of racist opinions in a sample of high school students.

The study design is well conducted. The sample, although local, is appropriate to the objectives of the study.

The questionnaires are also appropiate to these objectives.

The authors correctly warn of the limitations of the study, derived from the local nature of the sample and the cross-sectional nature of the study.

There are some recommendations that I think could help improve the content of the paper:

1. Table 1 includes too many information. It's suggested to divide it in at least two different tables.
Authors should consider to relocate the description of participants (number, sex, race and school course and academic performance; questions 1-5) from results' section to methods' section (adding a new participants' subsection).
Also, it could be considered to divide the rest of the table in two different tables, one of them regarding to online activity (questions 6, 8-16), and another one regarding results of the study (questions 7, 17-19, 22).

2. In the discussion section the first paragraph should be reduced in its length. It repeats some information included in the introduction.
Maybe just briefly recalling the objectives would be a good way to start the discussion without repeating too much information from the introduction

3. At the end of the discussion, authors might consider briefly including some recommendations for the training of young people (or its teachers and families) in the identification and handling of dangerous situations in online environments related to hateful messages and behaviors.

Finally, as just a formal issue, the letter p indicating the probability value must be in italics.

Author Response

The article presents the results of an interesting study that analyzed exposure to hate messages and the search for someone who tries to convince the respondent of racist opinions in a sample of high school students. The study design is well conducted. The sample, although local, is appropriate to the objectives of the study. The questionnaires are also appropriate to these objectives. The authors correctly warn of the limitations of the study, derived from the local nature of the sample and the cross-sectional nature of the study. There are some recommendations that I think could help improve the content of the paper:

Point 1: Table 1 includes too many information. It's suggested to divide it in at least two different tables.

Response 1: Thank you for this helpful feedback, we have divided Table 1 into three tables:

1) Questions regarding online activity (q6, 8-16)

2) Summary statistics (mean, standard deviation, median, and range) for the scales (q7, q19)

3) Questions corresponding to the outcomes of the study (q17, q22)

Point 2: Authors should consider to relocate the description of participants (number, sex, race and school course and academic performance; questions 1-5) from results' section to methods' section (adding a new participants' subsection).

Response 2: Thank you for highlighting this issue – we have moved questions 1-5 from the results section (and removed them from Table 1) to the methods section, adding a new participants section (2.1). You will find these changes beginning on lines 189.

Point 3: Also, it could be considered to divide the rest of the table in two different tables, one of them regarding to online activity (questions 6, 8-16), and another one regarding results of the study (questions 7, 17-19, 22).

Response 3: Please see our response to Point 1. Thank you for this feedback, we believe that it improves the readability of the information provided previously in Table 1.

Point 4: In the discussion section the first paragraph should be reduced in its length. It repeats some information included in the introduction. Maybe just briefly recalling the objectives would be a good way to start the discussion without repeating too much information from the introduction

Response 4: We agree that this section was redundant and could be removed from the beginning of the discussion. Reviewer 1 brought this to our attention as well, and recommended we move some of the material from it to the introduction. The section is now reduced in length, and the study objectives are recalled – you will find them on lines 449

Point 5: At the end of the discussion, authors might consider briefly including some recommendations for the training of young people (or its teachers and families) in the identification and handling of dangerous situations in online environments related to hateful messages and behaviors.

Response 5: Thank you for bringing this perspective to the paper, we believe that, based on current research, influential adults in the lives of youth (parents and teachers) also play a large role in improving their online safety and digital literacy. We have added a paragraph to reflect this stance and suggest that efforts to improve youth online safety should also incorporate components aimed towards influential adults. You will find this paragraph on lines 535-545.

Point 6: Finally, as just a formal issue, the letter p indicating the probability value must be in italics.

Response 6: Thank you for this comment, we have italicized all the p-values in the manuscript.

Reviewer 3 Report

The subject of the work is interesting and current. It is missing in the introduction to mention other recent research on the subject. The structure is correct as well as the methodological approach. The results are well exposed and the conclusions are solid, although the initial statement of objectives to be contrasted is not appreciated. The bibliography is spot on.

It is recommended to advance in the establishment of the objectives of the investigation and the initial state of the question (previous investigations).

Author Response

Point 1: The subject of the work is interesting and current. It is missing in the introduction to mention other recent research on the subject.

Response 1: Thank you for your very helpful feedback! We have added a literature review section on the previously investigations into the risk factors of exposure to hateful materials online to the introduction – beginning on line 60. At the suggestion of reviewer 1 & 2, we moved some of the beginning of the conclusion to the introduction as part of this literature review – you will find these parts on lines 61 and 100.

Point 1: The structure is correct as well as the methodological approach. The results are well exposed and the conclusions are solid, although the initial statement of objectives to be contrasted is not appreciated. The bibliography is spot on.

Response 2: We have also included the objectives of the present manuscript – you will find them on line 169. At the beginning of the conclusion, we re-state these objectives so they may be contrasted – you will find this on line 449.

Point 3: It is recommended to advance in the establishment of the objectives of the investigation and the initial state of the question (previous investigations).

Response 3: As mentioned above we have added a new literature review section (line 60) and objectives (line 169) to the introduction.

Again, we do thank you for your very helpful feedback.

Round 2

Reviewer 1 Report

Dear authors,
I have read with great interest your revised manuscript and I think you did a good job!

Although some parts may still be improved, my idea is that the paper can be published.

Below I will list some comments / suggestions that you could add or make:

  • In line 114 you could indicate that the questionnaire is quantitative (eg quantitative questionnaire) or add in the sentence that the methodology you apply is quantitative. Thus, you will be clear about the methodology method you used.
  • Also, in the introductory paragraph (lines 114-128) of section 2 (Materials and Methods) or any other point in this section where you are considered best you should write that your results will be presented (and) descriptively.
  • In section 4 (discussion) try where you can report your results, eg on lines 377-379 "While our results indicated that Twitter and Houseparty were associated with exposure to hate online, and YouTube was associated with exposure to an individual trying to convince the respondent of racist views (Table 3), [...] ".
  • In addition it should be mentioned / written that the results can not be generalized to the population, due to eg the small sample or anything else that is considered to be reported.
  • Also, section 4 could be renamed "Discussion and Conclusion" since you have removed the section "conclusions" after reviewing your manuscript.
  • Finally, check your bibliography (at the end), including its numbering. The manuscript I see seems to have a problem.

Author Response

Thank you for your very insightful and thoughtful feedback in the first round of revisions, we sincerely believe they greatly improved the manuscript!

We have accepted all the changes made in the first round of revisions so that you may easily see the changes in the second round. Please see the attached document for a point-by-point response of your feedback.
